# Embedding biocatalysts in a redox polymer enhances the performance of dye-sensitized photocathodes in bias-free photoelectrochemical water splitting

Fangwen Cheng[1], Olha Pavliuk [2], Steffen Hardt[3], Leigh Anna Hunt[1], Bin Cai[1], Tomas Kubart [4], Leif Hammarström [1], Nicolas Plumeré [5] ✉, Gustav Berggren [2] ✉ & Haining Tian [1] ✉

Dye-sensitized photoelectrodes consisting of photosensitizers and molecular catalysts with tunable structures and adjustable energy levels are attractive for low-cost and eco-friendly solar-assisted synthesis of energy rich products. Despite these advantages, dye-sensitized NiO photocathodes suffer from severe electron-hole recombination and facile molecule detachment, limiting photocurrent and stability in photoelectrochemical water-splitting devices. In this work, we develop an efficient and robust biohybrid dye-sensitized NiO photocathode, in which the intermolecular charge transfer is enhanced by a redox polymer. Owing to efficient assisted electron transfer from the dye to the catalyst, the biohybrid NiO photocathode showed a satisfactory photocurrent of 141±17 μA·cm$^{-2}$ at neutral pH at 0 V versus reversible hydrogen electrode and a stable continuous output within 5 h. This photocathode is capable of driving overall water splitting in combination with a bismuth vanadate photoanode, showing distinguished solar-to-hydrogen efficiency among all reported water-splitting devices based on dye-sensitized photocathodes. These findings demonstrate the opportunity of building green biohybrid systems for artificial synthesis of solar fuels.

The escalating problems with global warming emphasize the significance of building sustainable and scalable systems for energy conversion. Photo-assisted water splitting that can convert solar energy into hydrogen is a promising way to generate clean and renewable fuels[1]. Generally, photoelectrochemical (PEC) cells for water-splitting are based on a photoanode and a photocathode in a tandem configuration to obtain solar-driven hydrogen production without applied potential[2]. Dye-sensitized photoelectrodes with eco-friendly materials and adjustable energy levels have broad applications in PEC water splitting[3,4], $CO_2$ reduction[5,6], and organic fuel synthesis[7]. In a dye-sensitized PEC water-splitting system, the overall device performance is currently limited by the cathodic side[8]. While the optimal dye-sensitized photoanodes[9,10] can generate photocurrents beyond 1 mA·cm$^{-2}$, the stabilized photocurrents of dye-sensitized photocathodes[11–15] with co-sensitized catalyst or covalently linked catalyst were mostly limited to less than 100 μA·cm$^{-2}$. To date, most

[1]Department of Chemistry—Ångström laboratory, Physical Chemistry, Uppsala University, Box 521, 75120 Uppsala, Sweden. [2]Department of Chemistry—Ångström laboratory, Molecular Biomimetics, Uppsala University, Box 523, 75120 Uppsala, Sweden. [3]Institute of Energy and Climate Research, Fundamental Electrochemistry (IEK-9), Forschungszentrum Jülich GmbH, Wilhelm-Johnen-Straße, 52425 Jülich, Germany. [4]Department of Electrical Engineering, Solid-State Electronics, Uppsala University, Box 65, 75103 Uppsala, Sweden. [5]TUM Campus Straubing for Biotechnology and Sustainability, Technical University of Munich, Uferstrasse 53, 94315 Straubing, Germany. ✉e-mail: nicolas.plumere@tum.de; gustav.berggren@kemi.uu.se; haining.tian@kemi.uu.se

solar-to-hydrogen (STH) efficiencies in the few reported overall water-splitting PEC cells based on dye-sensitized photocathode only achieved a value less than 0.005%[15–17], far from the threshold needed for applications (>10%)[18]. Recently, an STH efficiency of 0.11% from a dye-sensitized PEC water-splitting device has been reported, but the photocurrent was dropped significantly after 15 min light irradiation[8]. The insufficient photocurrents as well as the poor stability of the photocathode, hinder the progress of dye-sensitized water splitting.

In dye-sensitized NiO photocathodes, although photoinduced hole injection takes place at femtosecond level from surface-bound photoexcited dye molecules to mesoporous NiO[19,20], the undesired charge recombination between the reduced dye and the valence band or surface states of NiO is also fast, normally at picosecond timescale[21–23]. In the presence of a catalyst on the NiO photocathode, the electron transfer from the reduced dye to the catalyst needs to compete with the aforementioned charge recombination step and is, therefore, a crucial step for the subsequent catalytic reactions. Many efforts have been made to facilitate the electron transfer process to the catalyst, for example, optimizing the molecular structures of the photosensitizers and catalysts[24] and combining the catalysts with the photosensitizers such as, simply drop-casting[25,26], layered adsorption[27,28], co-sensitization[14,29] and covalently linking the catalysts with the photosensitizers[13,19,30–32]. Nevertheless, charge recombination remains excessive in dye-sensitized photocathodes, making cathodic photocurrents unable to match the state-of-the-art anodic photocurrents to complete an efficient overall water splitting. We recently demonstrated that the use of ZnO and TiO2 as electron transport materials to mediate the charge transfer from the dye to the Pt as the catalyst, leading to a significantly improved and stabilized photocurrent of 100 µA cm$^{-2}$ at pH 5 with a bias potential at 0.05 V vs. RHE[33]. However, fabrication of such a NiO photocathode requires a precious metal catalyst (Pt) and energy-consuming steps such as atomic layer deposition plus thermal sintering to prepare electron transport layers. Hydrogenase, a noble-metal-free catalyst, has shown high catalytic activity for hydrogen production with energy efficiencies comparable to Pt catalysts[34], and therefore is a good candidate for dye-sensitized NiO photocathode in PEC devices.

In this work, we developed a biohybrid dye-sensitized NiO photocathode in which the organic dye PB6 acts as the photosensitizer, the [FeFe] hydrogenase (H2ase) serves as a biological catalyst for proton reduction, and the viologen-based redox-active polymer (PolyV) functions as an electron mediator. Fast electron transfer from PB6 to PolyV suppresses charge recombination. The subsequent electron transfer from the reduced PolyV to H2ases enables efficient H2 production. Moreover, immobilization of H2ases in the PolyV matrix can stabilize H2ases on the photocathode by retarding the detachment of the H2ases. The developed biohybrid photocathode showed a photocurrent of 141 ± 17 µA·cm$^{-2}$ at pH = 7 under 0 V vs. RHE with a stable continuous output for 5 h. The photocurrent and stability are superior to all reported dye-sensitized photocathodes with molecular catalysts. Furthermore, the designed photocathode was used to construct a bias-free overall water-splitting PEC cell, showing a photocurrent of 60–70 µA·cm$^{-2}$ with only a slight decline during 10 h of continuous operation, rendering a solar-to-hydrogen efficiency of 0.124% (calculated with irradiance of 50 mW·cm$^{-2}$) that also outperforms all full water-splitting devices based on dye-sensitized molecular photocathodes (Table 1).

## Results
### Biohybrid photocathodes
The energy levels of each component in the photocathode are depicted in Fig. 1a. A compact NiO-blocking layer underneath a mesoporous NiO film is employed to avoid direct contact between the catalyst/redox polymer and the conducting glass substrate (fluorine-doped tin oxide, FTO). PB6, with the donor-π-acceptor structure shown in Fig. 1b,

is a purely organic molecular photosensitizer. As demonstrated in our previous studies[35], the oxidation potential of PB6 (1.03 V vs. normal hydrogen electrode (NHE) at pH = 7) thermodynamically matches with the valence band (VB) of mesoporous NiO (0.50 V vs. NHE), which enables efficient electron injection from NiO to excited PB6[36]. The highly active [FeFe]-H2ase from *Chlamydomonas reinhardtii* (*Cr*HydA1) was mixed with 2,2′-viologen-based redox polymer (PolyV, Fig. 1b) and immobilized on the PB6-sensitized NiO photoelectrode by drop-casting. The reduction potential for PolyV is determined to be −0.59 V vs. NHE at pH = 7 (Supplementary Fig. 1), which is close to the reduction potential of the H2ase-catalyst and the proton/dihydrogen couple at neutral pH[37], while more positive than the reduction potential of PB6 dye (−0.93 V vs. NHE)[35]. Previously, we have demonstrated that PolyV enables efficient electron transfer to [FeFe]-H2ase on electrode surfaces, enabling reversible proton reduction catalysis with negligible overpotential and 100% Faradaic efficiency[38]. The use of PolyV as a fast electron mediator in a dye-sensitized photocathode is designed to suppress the charge recombination by enabling fast electron transfer towards the H2ase, which is the prerequisite for efficient solar-to-hydrogen energy conversion.

The structure of this biohybrid dye-sensitized photocathode was studied by cross-sectional scanning electron microscope (SEM) and energy-dispersive X-ray (EDX) spectra, as shown in Fig. 2 and Supplementary Fig. 2. As compared with the information from bare PB6 adsorbing on NiO (Supplementary Fig. 3), the distribution of Fe, Cl and Br elements in NiO|PB6|PolyV|H2ase sample demonstrates that both H2ase and PolyV penetrate well into nanoporous NiO. The intimate physical contact between each component in this photocathode enables the possibility of efficient intermolecular charge transfer for the hydrogen evolution reaction.

### PEC performance for H2 production
The PEC performance of the photocathode was investigated by linear sweep voltammetry (LSV) measured in aqueous Tris buffer (pH = 7), and the cathodic photocurrents were compared at 0 V vs. reversible hydrogen electrode (RHE). In order to obtain the best performance for photo-assisted H2 production, the assembly of the photoelectrode was first optimized. The optimal thickness of nanoporous NiO is determined to be 1 µm (Supplementary Figs. 4–6), and the optimal concentration of the 5 µL H2ase deposited onto the electrode is 50 µmol·L$^{-1}$ (Supplementary Fig. 7) to generate the highest photocurrent. The addition of 5 µL PolyV aqueous solution resulted in significantly enhanced photocurrent (Fig. 3a). The photocurrent increased with higher PolyV concentration and reached a plateau with 3 mg·mL$^{-1}$ PolyV. Moreover, the comparison of LSV plots for NiO|PB6|PolyV|H2ase with or without compact NiO (Supplementary Fig. 8) demonstrated the necessity of an electron-blocking layer, which can prevent the charge recombination from the catalysts to the FTO substrate, which differs from other dye-sensitized systems with small-molecular catalysts binding directly onto nanoporous NiO or the chromophores[39].

The optimal LSV curves for each electrode in the presence of a compact NiO layer measured under chopped light are shown in Fig. 3b. The small background photo response for the PB6-sensitized NiO sample can be ascribed to proton reduction on the NiO surface due to electron back transfer from the PB6 to surface states of NiO[36]. The obvious positive transient currents largely result from the charge accumulation at the surface states on NiO upon light illumination. Simply adding PolyV (NiO|PB6|PolyV) did not make a large difference to the photocurrent, but the addition of H2ase (NiO|PB6|H2ase) showed enhanced cathodic photocurrent. Integration of both PolyV and H2ase on the photocathode (NiO|PB6|PolyV|H2ase) resulted in the highest photocurrent among four photoelectrodes, reaching up to 141 ± 17 µA·cm$^{-2}$ photocurrent under 0 V vs. RHE. Notably, the LSV scanning of NiO|PB6|PolyV|H2ase showed reduced negative transient

**Table 1 | PEC performance for non-assisted overall water splitting using a molecular-based p-type dye-sensitized photocathode**

| Structure | Stabilized photocurrent density | Faradaic efficiency (H$_2$) | STH efficiency | Ref. |
|---|---|---|---|---|
| NiO\|PMI-6T-TPA\|\|BiVO$_4$ | 2 µA·cm$^{-2}$ (4 h) | 80% | – | 16 |
| NiO\|RuP\|CoHEC\|\|TiO$_2$\|RuP\|RuOEC | 12 µA·cm$^{-2}$ (120 s) | – | – | 42 |
| NiO\|P1\|Co1\|\|TiO$_2$\|L0\|Ru1 | 50 µA·cm$^{-2}$ (600 s) | 55% | 0.0051% | 15 |
| CuGaO$_2$\|RBG174\|CoHEC\|\|TaON\|CoO$_x$ | 4.5 µA·cm$^{-2}$ (3 h) | 87% | 0.0054% | 17 |
| NiO\|QAP-C8\|Co2\|\|TiO$_2$\|QAP-C8\|Zr$^{4+}$\|RuOEC | 58 µA·cm$^{-2}$ (900 s) | 89% | 0.11% | 8 |
| NiO\|PB6\|PolyV\|H$_2$ase\|\|BiVO$_4$ | 63 µA·cm$^{-2}$ (10 h) | 81% | 0.124% | This work |

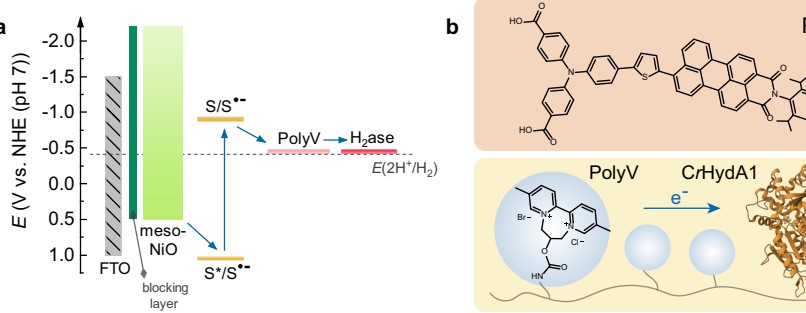

**Fig. 1 | Structure diagram of the biohybrid dye-sensitized photocathode.**
**a** Energy diagram for the photocathode and **b** molecular structures of PB6, PolyV, and H$_2$ase (*Cr*HydA1). The blocking layer is a compact NiO-blocking layer. S is short for the ground state of PB6, S* stands for excited PB6, and S$^{•-}$ is for reduced PB6. The blue arrows show the preferred electron transfer pathway.

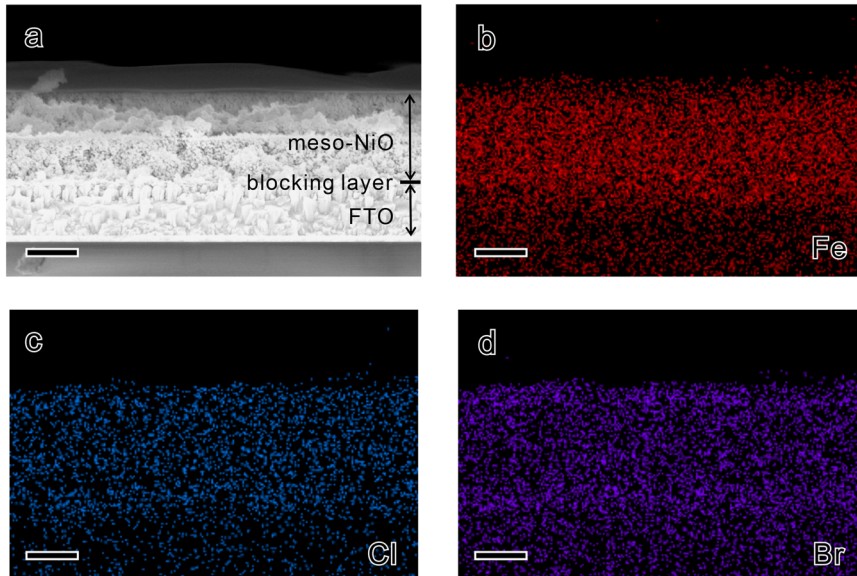

**Fig. 2 | Characterizations for the vertical structure of the photocathode. a** Cross-sectional SEM images for NiO\|PB6\|PolyV\|H$_2$ase photocathode. **b–d** EDX images for Fe, Cl, and Br. The scale bar is 500 nm.

spike current and no positive transient current, indicating that the electron electron-hole recombination has been significantly suppressed. NiO\|PB6\|PolyV\|H$_2$ase exhibited increased incident photon-to-current efficiencies (IPCE) compared to the one without PolyV with same absorption of PB6, indicating assisted charge transfer by the redox polymer (Supplementary Fig. 9). Moreover, the generated photocurrents are similar in electrolytes with a pH range from 6 to 7.5 (Supplementary Fig. 10). This pH independence of the photocurrent shows that electron transfer between PolyV and H$_2$ase, which has been shown to correlate with pH[38], is not the limiting step for the PEC performance.

Dye-sensitized NiO photocathodes bear instability mainly due to the easy detachment of small-molecular catalysts. The direct combination of H$_2$ases and dye-sensitized NiO substrates showed fast decay in the photocurrent within the first hour of output (Fig. 3c), similar to previously reported H$_2$ase-based biohybrid photocathodes[40]. Possible instability of PB6 was excluded, because of its unchanged absorption after 5 h of operation (Supplementary Fig. 11). The H$_2$-evolution assay showed only 37% of the remaining HER activity of the enzyme after the experiment, either caused by enzyme detachment or deactivation. This could be effectively mitigated by incorporating the enzyme into PolyV. With the addition of PolyV, the biohybrid electrode had 78% of

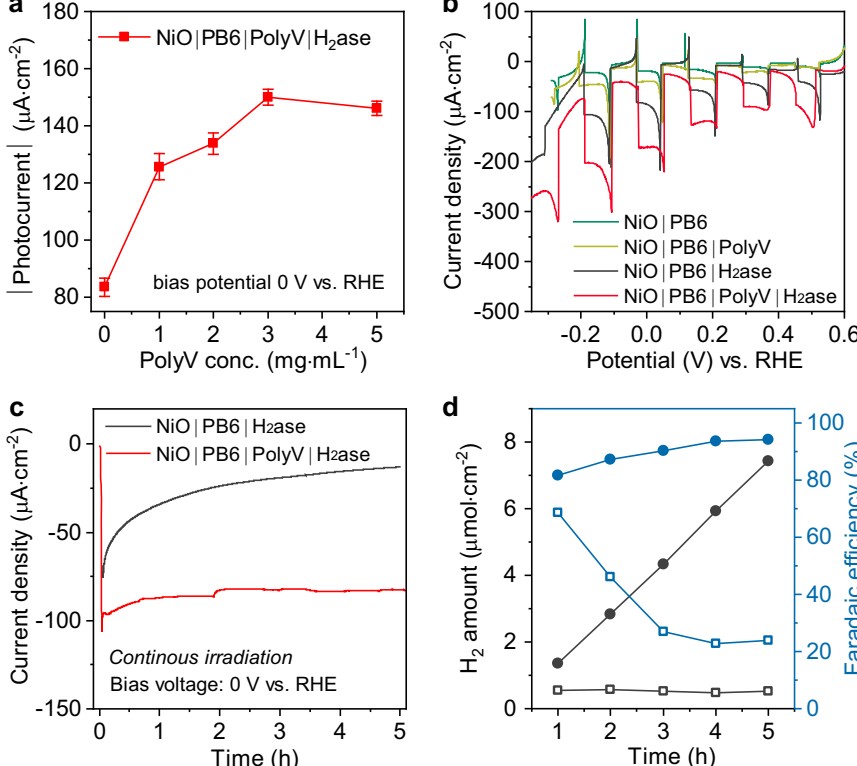

**Fig. 3 | Catalytic performance of the photocathodes. a** The photocurrent density at 0 V vs. RHE for NiO|PB6|PolyV|H₂ase with different PolyV amounts. Error bars show standard deviations between the three films. **b** LSV curves with chopped light. **c** Chronoamperometric measurements of the photocathodes recorded at an applied potential of 0 V vs. RHE. **d** Measurements of generated H₂ amount and Faradaic efficiency for H₂ production at 0 V vs. RHE. The black lines are used for the H₂ amount and the blue lines are for the Faradaic efficiency. The hollow squares referred to NiO|PB6|H₂ase and the solid circles specified for NiO|PB6|PolyV|H₂ase.

the enzymes remaining active after the reaction. The observation that high photocurrents were sustained over hours despite a loss of about 20% in catalyst loading is in line with charge recombination at the NiO surface being the main limiting factor. Indeed, the resulting NiO|PB6|PolyV|H₂ase cathode exhibited benchmark stability for photocurrent during continuous output for 5 h, outperforming all previously reported dye-sensitized photocathodes with small-molecular catalysts (Supplementary Table 2). The results prove that the redox polymer can effectively immobilize and protect H₂ase on the electrode surface and thus improve the stability of the photocathode. The Faradaic efficiency for photo-assisted H₂ production was determined by measuring the amount of H₂ produced in the headspace of the PEC cell. After 5 h of continuous light irradiation at 0 V vs. RHE, 7.4 μmol·cm⁻² H₂ was detected, corresponding to a Faradaic efficiency of 94% for the NiO|PB6|PolyV|H₂ase electrode assuming completely leak-free conditions. The NiO|PB6|H₂ase electrode exhibited 68% Faradaic efficiency in the first hour, but Faradaic efficiency diminished quickly due to enzyme detachment and/or deactivation. The high Faradaic efficiency of the photocathode in the presence of PolyV can be assigned to more efficient electron extraction to H₂ase and less charge recombination. The PEC performance, including the photocurrent, stability, and generated H₂ amount, reported in this work stands out among the reported NiO-based photocathodes (Supplementary Table 2), which inspired us to study the electron transfer kinetic mechanism in this electrode.

### Mediated electron transfer by redox polymer

To rationalize the improvement in the PEC device's photocurrent, time-resolved spectroscopy was conducted to investigate the dynamics of the electron transfer processes at the photocathode. From the pH dependence test, it was determined that electron transfer between PolyV and H₂ase is not limiting the device's photocurrent.

However, both the electron and hole transfer processes from PB6 are crucial steps that could potentially be device-limiting. Thus, time-correlated single photon counting (TCSPC) was employed to investigate the electron/hole injection rate from excited PB6. As shown in Supplementary Fig. 12, the PB6 photoluminescence in NiO|PB6 decays within 200 ps, indicating hole injection from excited PB6 to NiO, which is significantly faster than electron transfer from excited PB6 to either H₂ase or PolyV/H₂ase. This is consistent with previously reported results that photoexcitation of NiO|PB6 is followed by ultrafast hole injection into NiO within 200 fs[35] to generate reduced dye (PB6•⁻). Given the suggested timescale of PB6•⁻ formation, femtosecond transient absorption (fsTA) experiments were carried out to monitor the subsequent charge transfer steps.

In the NiO|PB6 film, the fsTA data after 560 nm excitation is in very good agreement with previous reports[35]. The initial TA spectra show positive absorption bands at 420 and 680 nm, and two characteristic PB6 ground state bleach regions at 360 and 530 nm (Fig. 4a). Within 0.4 ps, the 680 nm band shifts to 650 nm, and the isosbestic point around 600 nm blueshifts. The initial spectral features are attributed to a mixture of excited PB6 and the reduced species PB6•⁻, based on a comparison of fsTA data for excited PB6 in ZrO₂|PB6 samples and spectroelectrochemistry data of PB6•⁻ [35]. PB6•⁻ forms after ultrafast hole injection from excited PB6 into NiO, its signal at 650 nm maximizing after ca. 0.4 ps, which is followed by decay to the ground state via charge recombination on time scales of 10's of ps to ns. The fsTA data were globally analyzed with a sum of three exponentials. Multiexponential hole injection and recombination is commonly reported for NiO|dye systems; however, given the heterogeneous nature of the decay kinetics, the time scales of the underlying charge transfer processes are largely approximations, and reported time constants are likely to include overlapping charge transfer processes. Global analysis

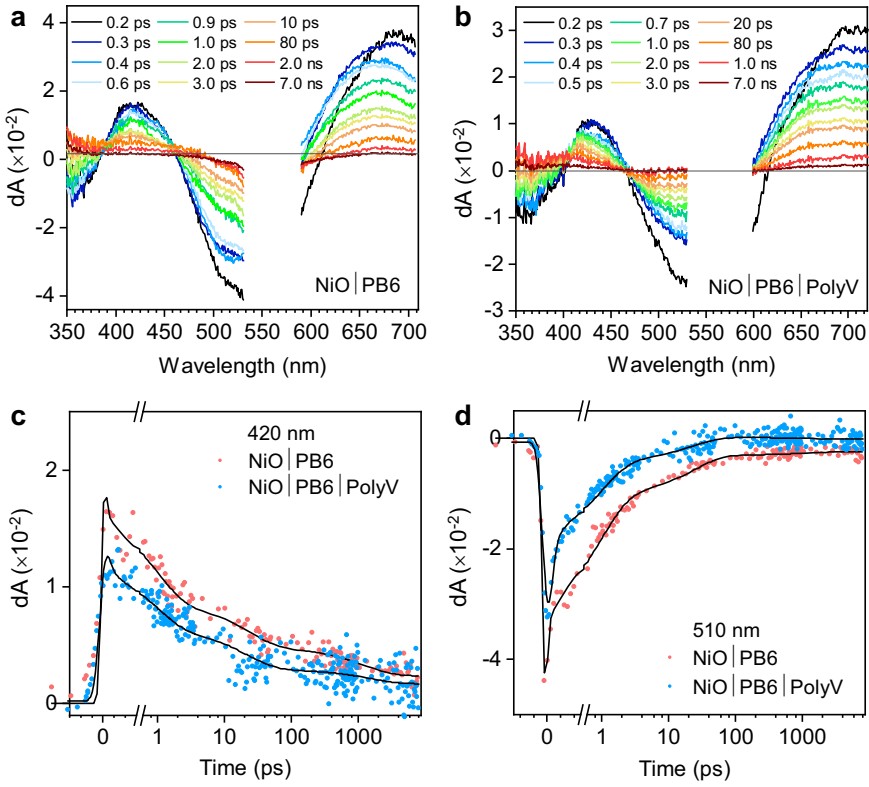

**Fig. 4 | Ultrafast transient spectroscopy of the photocathodes. a, b** fsTA spectra at selected delay times for NiO|PB6 film and NiO|PB6|PolyV film following excitation at 560 nm. **c, d** Single-wavelength kinetics at 420 and 510 nm for both samples.

of the NiO|PB6 data resulted in the following lifetimes: $\tau_1 = 0.9$ ps, $\tau_2 = 20$ ps, and $\tau_3 = 1.3$ ns (Supplementary Fig. 14). The first time component ($\tau_1$) is assigned to hole injection from excited PB6 to NiO while the second ($\tau_2$) and third ($\tau_3$) time components are attributed to multiphasic charge recombination between PB6$^{\bullet-}$ and NiO(+), which is in good agreement with our previous work[27,35].

When PolyV is added to the system, the initial spectral features observed are similar to those observed for the NiO|PB6 film (Fig. 4b). However, while the absorption features of the NiO|PB6 film decay without any notable spectral evolution, the 430 nm absorption band in the NiO|PB6|PolyV film blueshifts and becomes narrower after the first 0.3 ps. The observed spectral contribution can be explained with the assumption of electron transfer to PolyV to form reduced PolyV, which is in good agreement with spectroelectrochemical measurements that show reduced PolyV absorption peak at roughly 400 nm (Supplementary Fig. 13). The reduction of PolyV can happen directly, by electron transfer from PB6$^{\bullet-}$, or indirectly, following a surface state-mediated pathway, where NiO surface states are first reduced by PB6$^{\bullet-}$ before donating electrons to PolyV[36]. The resulting time constants from global analysis of the NiO|PB6|PolyV data are as follows: $\tau_1 = 1.0$ ps, $\tau_2 = 25$ ps, and $\tau_3 = 3.2$ ns, where the first time component ($\tau_1$) is again assigned to hole injection from excited PB6 to NiO. The evolution-associated decay spectra (Supplementary Fig. 15) suggest that the fast component mainly forms NiO$^+$-PB6$^{\bullet-}$ and that the formation of reduced PolyV by electron transfer from PB6$^{\bullet-}$ or reduced NiO surface states is represented by the 25 ps ($\tau_2$) component. The third ($\tau_3$) time component can be described as a convoluted mixture of the long-lived decay of reduced PolyV and charge recombination between PB6$^{\bullet-}$ and NiO(+). Additionally, the third time constant ($\tau_3 = 3.2$ ns) is slower in comparison to that observed in the NiO|PB6 film. This component has a more significant contribution in the 420 nm kinetics (reduced PolyV, Fig. 4c), but very little contribution at 510 nm (PB6 recovery, Fig. 4d) and 680 nm decay (PB6$^{\bullet-}$, Supplementary Fig. 16).

This suggests a slower charge recombination phase between reduced PolyV and NiO(+), compared to the recombination between PB6$^{\bullet-}$ and NiO(+) in NiO|PB6.

Overall, the fsTA results suggest that following ultrafast hole injection to form PB6$^{\bullet-}$, PolyV can efficiently accept electrons from either PB6$^{\bullet-}$ or reduced NiO surface states to form reduced PolyV. Thus, PolyV is a suitable redox polymer to rapidly recover electrons from the dye-sensitized photocathode. The rapid formation and long-lived reduced state of PolyV is likely also useful for preventing charge recombination and enabling efficient electron transfer to H$_2$ase at the dye-catalyst interface, which contributes to the enhanced catalytic performance of the biohybrid dye-sensitized photocathode.

### Bias-free overall water-splitting PEC cells
The high-performance biohybrid dye-sensitized NiO photocathode provides the possibility of fabricating a bias-free overall solar water-splitting device. For the photoanode side, BiVO$_4$ with FeOOH/NiOOH cocatalysts was chosen due to its high activity[41] and matched energy level (Fig. 5a). Three-electrode PEC analysis of the BiVO$_4$ photoanode gave an onset potential of +0.13 V vs. RHE and a current density of 0.75 mA cm$^{-2}$ at +0.4 V vs. RHE (Fig. 5b). The comparison of the three-electrode LSV scans between the photocathode and the photoanode showed an intersectional photocurrent of approximately 100 μA cm$^{-2}$, suggesting the feasibility of unassisted water splitting in a tandem structure and indicating that the photocurrent of the tandem device was limited by the photocathode. A two-electrode configuration with a Nafion membrane separating the anodic and cathodic compartments was adapted to test the bias-free water-splitting performance (the inset of Fig. 5d). Chronoamperometric scans with irradiation from each side were obtained without applied potential (Fig. 5c). When the tandem cell was illuminated from the cathodic side, it showed a stable photocurrent of $91 \pm 8$ μA cm$^{-2}$, whereas illumination from the anodic side resulted in much inferior performance exclusively showing capacitive

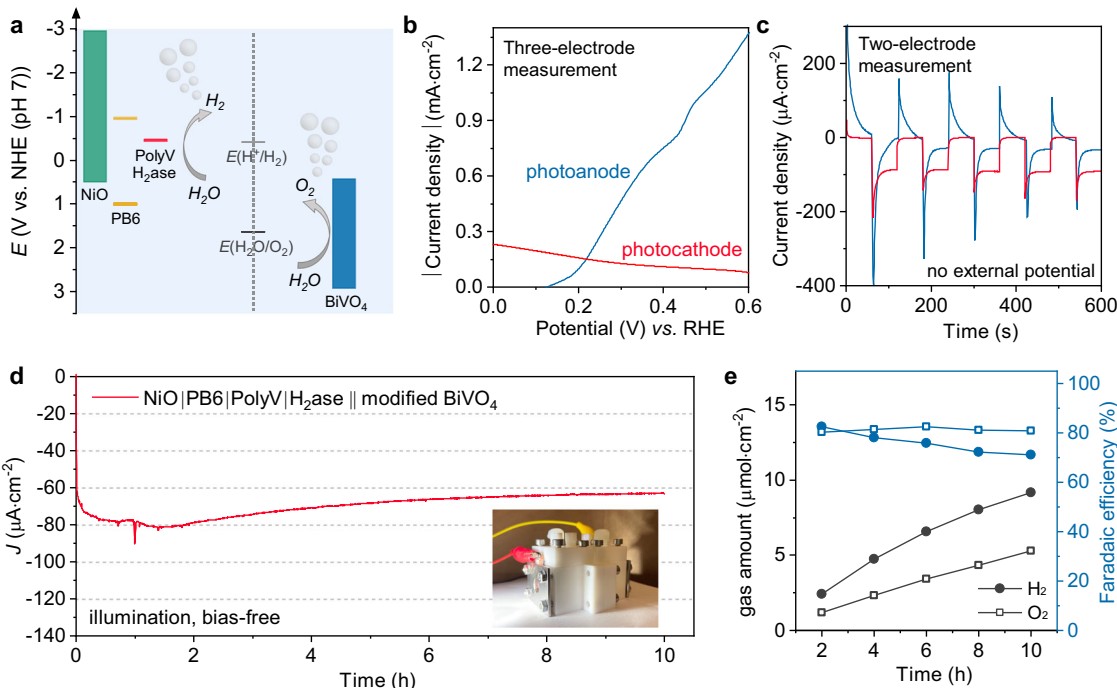

**Fig. 5 | Performance of the overall water splitting system. a** Scheme of the device for overall water spitting. **b** LSV scans of NiO|PB6|PolyV|H₂ase (photocathode) and modified BiVO₄ (photoanode) under illumination, obtained from three-electrode measurements. The photocurrent density of the photocathode was inverted for comparison. **c** Chronoamperometric plots for bias-free tandem PEC cells under chopped illumination from the anodic side (blue) or the cathodic side (red).

**d** Stability of bias-free tandem water-splitting PEC cells under cathodic illumination with a light intensity of 50 mW·cm⁻² (420–750 nm). The inset shows the photo of the two-electrode tandem device. **e** Generated H₂ and O₂ amount and Faradaic efficiency during bias-free water splitting under cathodic illumination. The black lines are for the gas amount and the blue lines are for the Faradaic efficiency. The solid circles are referred to as H₂ and the hollow squares referred to as O₂.

charging of the photocathode due to the light fully blocked by BiVO₄ electrode (Supplementary Fig. 17).

The long-term stability test of the tandem water-splitting device was tracked with irradiation from the cathodic side. The tandem PEC cell exhibited a stable photocurrent of 63 μA·cm⁻² during constant output (Fig. 5d). Three-electrode LSV scans for both the anode and the cathode before and after the reaction showed that the slight photo-current decay is mainly due to the performance reduction of the photocathode (Supplementary Fig. 18), probably due to small amount of H₂ase detached or deactivated on the surface. The amounts of H₂ and O₂ gas generated in the headspace of the cathode and the anode side, respectively, were tested every 2 h (Fig. 5e). The ratio between H₂ and O₂ was approaching 2:1. Faradaic efficiencies of H₂ and O₂ pro-duction both reached over 80% at the first 2-h measurement point. Faradaic efficiency for O₂ kept stable for 10 h while the value for H₂ decreased slightly. The STH efficiency of the bias-free tandem device was determined to be 0.124% from a 10-h reaction. The performance and stability of this tandem device shows a breakthrough in reported overall PEC water-splitting systems using molecular-based dye-sensi-tized photocathodes (Table 1).

## Discussion

An efficient and robust biohybrid photocathode has been achieved by integrating a viologen-based redox-active polymer and [FeFe]-hydro-genase onto a PB6-sensitized NiO electrode. The transient spectro-scopy data have shown that the redox polymer enables efficient electron extraction from the reduced dye and thus bypasses charge recombination at the photoelectrode. The redox-active polymer sub-sequently transfers the electron to the catalyst for efficient H₂ pro-duction. Moreover, the PEC stability was significantly enhanced by the introduction of the redox-active polymer used to immobilize, wire, and protect the H₂ase on the photocathode. The high-performing photocathode is compatible with BiVO₄ to fabricate bias-free tandem

PEC devices with benchmark performance, paving the way to tandem systems with biohybrid electrodes on both sides. The polymer design methodology provides a pathway for efficient bias-free photoelec-trochemical hydrogen synthesis, based on earth-abundant elements that could fulfill the scalability requirement.

## Methods

### Preparation and H₂-production assay of H₂ase (CrHydA1)

*Cr*HydA1 (Mw 49.43 kDa) was prepared following a reported protocol[37], with details further stated in the supplementary informa-tion. To test enzyme activity, the holo-enzyme after reconstitution and activation steps was diluted to a final concentration of 1 μM in 0.1 M phosphate buffer (pH = 6.8) containing 10 mM methyl viologen (final concentration) in an 800 μL gas-tight vial. The reaction mixture was then initiated by the rapid addition of sodium dithionite (final con-centration 100 mM) to the final volume of 180 μL. The resulting mix-ture was incubated at 37 °C for 15 min. Following this incubation period, a 100 μL sample from the gas phase was introduced into a gas chromatograph (GC) (PerkinElmer Clarus 500). The enzymatic activity was calculated as 240 μmol H₂·min⁻¹·mg⁻¹ (TOF = 12,000 min⁻¹).

### Fabrication of biohybrid dye-sensitized photocathodes

The PB6-sensitized nanoporous NiO layers were prepared following the previous work in our group[35]. A 100 nm compact NiO layer was added between fluorine-doped tin oxide (FTO) glass and nanoporous NiO by reactive magnetron sputter deposition (Von Ardenne, CS730S). A mixture of PolyV (5 μL, H₂O, 3 mg·ml⁻¹) and *Cr*HydA1 (5 μL, 50 μM, Tris/HCl buffer, pH = 8) was immobilized by drop-casting on top of the sensitized NiO layer in the N₂ atmosphere. The optimized PolyV and *Cr*HydA1 concentrations were 3 mg·ml⁻¹ and 50 μM, respectively. After the mixture solution underwent adsorption for 15 min, any excess solution was removed from the surface. The thin-film absorption spectra were carried out with a spectrophotometer (Varian, Cary

5000). The cross-sectional structure of the photocathode was obtained by SEM (Zeiss, LEO 1550) equipped with Oxford AZtec EDX system.

## Spectroscopic measurements

For the comparison, mesoporous NiO films sensitized with PB6 were used from the same batch. TCSPC was carried out with a spectrofluorometer (Edinburgh FS5) equipped with a picosecond diode laser of 470.4 nm (EPL 470). The samples with H$_2$ase were sealed with epoxy adhesive (LOCTITE EA 9466) in the N$_2$ atmosphere. The fitting parameters for TCSPC were listed in Supplementary Table 1. Spectroelectrochemistry of PolyV film drop-casted on FTO was performed in Tris buffer (same as the electrolyte for PEC measurement) with a 1-cm path length cuvette. Three-electrode electrochemical experiments were carried out using an Autolab potentiostat (PGSTA302). Before the measurement, oxygen was removed from the cell by argon bubbling. Time-resolved UV-vis/NIR spectra of the PolyV films were recorded with a diode array spectrometer (Hewlett Packard, 8453) during the reduction of PolyV at −0.6 V vs. NHE.

Ultrafast transient absorption (fsTA) experiments were carried out using a Ti:sapphire-based amplifier with integrated oscillator and pump lasers. The laser fundamental (800 nm, 3 kHz) was split into a pump and probe by a beam splitter, which were directed toward the sample chamber (TAS, Newport Corporation). The pump beam at 560 nm was generated by an optical parametric amplifier (TOPAS NirUVis, Light Conversion). The continuum probe beam was generated by focusing a few mJ/pulse of the amplifier fundamental on a translating CaF$_2$ crystal. Prior to the sample cell, the pump was passed through a depolarizer and attenuated using a neutral density filter. Briefly, the probe supercontinuum was generated from a calcium fluoride plate, and its path was controlled by an optical delay ($t_{window} \leq 8$ ns), allowing the transient spectra at varying pump–probe delay times to be recorded on a silicon diode array (Newport, custommade). The measurements were performed with an average pump power of 5 mW. The sample films were mounted on a translation stage and moved continuously in the vertical direction to refresh sample volume and avoid sample degradation during the measurements.

All TA data was initially analyzed using SurfaceXplorer (Ultrafast Systems) for fitting the chirp to a third-order polynomial function. The transient absorption data were then globally fit to a sum of exponential functions, convoluted by a Gaussian function representing the instrument response function (IRF; FWHM 150 fs). Global analysis was performed by least-squares fitting using the R package TIMP and its GUI Glotaran. A sum of exponentials with frequency-dependent amplitudes were fitted to the transient data using a sequential scheme, yielding the evolution-associated decay spectra (EADS).

## Preparation of modified BiVO$_4$ photoanodes

The BiVO$_4$ films were prepared similarly to the reported procedures[41] with slight modifications. BiOI was electrodeposited onto clean FTO glass at −0.1 V vs. Ag/AgCl for 3 min and was annealed at 450 °C for 2 h with 0.2 M vanadyl acetylacetonate/DMSO precursor solution. Then FeOOH and NiOOH were photo-deposited on BiVO$_4$ sequentially.

## Assembly and performance examination of PEC cells

All PEC measurements were conducted in a gastight two-compartment PEC device (Dyenamo HT-holder) separated by a Nafion membrane, under strictly anaerobic conditions in an N$_2$ atmosphere. The active areas were confined to 0.3 cm$^2$. A Tris buffer (5 mM Tris, 5 mM HEPES, 5 mM sodium acetate, 5 mM MES, 0.1 M sodium sulfate, pH = 7.0) was used as the electrolyte solution. For three-electrode PEC measurements, Ag/AgCl (3 M NaCl) was employed as the reference electrode and Pt was used as the counter electrode. The light source was an LED PAR38 lamp (17 W, 50 mW·cm$^{-2}$, 420−750 nm) that had a similar light intensity between

420 and 750 nm compared to one-sun illumination. The electrochemistry tests were carried out with an Autolab potentiostat (PGSTAT10) controlled by Gpes software. Linear sweep voltammetry was scanned from 0 to −1 V (vs. Ag/AgCl) with a scan rate of 0.01 V s$^{-1}$. Chronoamperometry was recorded at a potential of −0.2 V (vs. Ag/AgCl) without a cooling system. The potentials were converted using the equation E (V versus RHE) = E (V versus Ag/AgCl) + 0.059 V × pH + 0.207 V at 298 K. Incident photon-to-current efficiency (IPCE) measurements were conducted in 5 mM Tris buffer under illumination of LED lamps with different wavelengths (470 nm, 15.1 mW·cm$^{-2}$; 590 nm, 2.5 mW·cm$^{-2}$; 630 nm, 8.9 mW·cm$^{-2}$). IPCE values were calculated from Eq. (1). The activity of the enzyme on the electrode surface before and after the 5-h reaction was analyzed by putting the electrode sample into 0.1 M phosphate buffer (pH = 6.8) containing 10 mM methyl viologen in a gas-tight vial and measuring the generated H$_2$ amount in GC (PerkinElmer Clarus 500) after incubation at 30 °C for 15 min following injection of sodium dithionite (final concentration 100 mM). The final volume of the solution was 400 µL. The percentage of residual enzyme activity after the reaction was obtained by the ratio of H$_2$ production. For overall water splitting, two-electrode measurements were conducted by connecting the photocathode and photoanode without any applied potential. The generated gas amounts were quantified by injecting 100 µL gas from the headspace (2.5 mL for both compartments) into the GC column. The Faradaic efficiency for H$_2$ and O$_2$ was calculated with equation (2). The solar-to-hydrogen (STH) efficiency was obtained from equation (3).

$$IPCE = \frac{1240 \times |J_{ph}|}{P_{mono} \times \lambda} \times 100\% \quad (1)$$

$$\text{Faradaic efficiency} = \frac{N \times e \times N_A \times n_{gas}}{Q} \times 100\% \quad (2)$$

$$\text{STH efficiency} = \frac{J_{ph}(\text{mA cm}^{-2}) \times (1.23V - V_{bias}) \times \text{Faradaic efficiency}}{P_{light}} \times 100\% \quad (3)$$

$J_{ph}$ is the photocurrent. $P_{mono}$ is the intensity of the single-wavelength (λ) light. N is the electron amount involved in the generation of one H$_2$/O$_2$ molecule (N = 2 for H$_2$, N = 4 for O$_2$), e is the elementary charge, N$_A$ is the Avogadro constant, n$_{gas}$ is the produced gas amount, Q is the amount of the passing charge. $V_{bias}$ is the bias voltage added to the two electrodes (0 V) and $P_{light}$ is the irradiance intensity (50 mW·cm$^{-2}$).

## Data availability

All data generated in this study are provided in the article and Supplementary Information. Source data are provided with this paper.

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

## Acknowledgements

We gratefully thank the financial support from Göran Gustafsson Foundation (to H.T.), The Swedish Energy Agency (STEM, grant no 48574-1 to G.B.), the Wenner-Gren Foundation (grant no GFU2022-0009 to O.P. and G.B.; grant no. 2021.0151 to H.T. and B.C.), the ANR-DFG "solar-driven chemistry" (grant no. PL746/5-1 to N.P.), the ERC Consolidator Grant E-VOLUTION (grant no. 101045008 to N. P.) and Knut and Alice Wallenberg Foundation (Grant 2019.0071 to L.H.) We also acknowledge Myfab Uppsala for providing facilities and experimental support. Myfab is funded by the Swedish Research Council (2019-00207) as a national research infrastructure. We appreciate the help from Princess Cabotaje for the verification test of enzyme activity.

## Author contributions

F.C. fabricated and characterized the devices and wrote the manuscript; O.P. synthesized the *Cr*HydA1 hydrogenase; S.H. synthesized the PolyV redox polymer; B.C. synthesized the PB6 dye; T.K. deposited compacted NiO layer; L.A.H. and L.H. worked on transient absorption experiments; N.P., G.B. and H.T. supervised and directed this project. All authors commented on the manuscript.

## Funding

## Competing interests

The authors declare no competing interests.
