## [Peer Review File · Nature Communications]

Embedding biocatalysts in a redox polymer enhances the performance of Dye-Sensitized Photocathodes in Bias-free Photoelectrochemical Water SplittingREVIEWER COMMENTS

Reviewer #1 (Remarks to the Author):

The manuscript by Tian and co-workers describes the preparation of a photocathode for hydrogen evolution based on sensitized NiO and incorporation of a hydrogenase in a redox-active viologen-based polymer. The photocathode is highly active towards photoelectrochemical hydrogen production reaching unprecedented photocurrent densities. Interestingly, the photocathode has been also combined with a BiVO₄ photoanode to perform bias-free photoelectrochemical water splitting. The manuscript is interesting and the results unprecedented in the field so that it may warrant publication. The following comments should be, however, addressed before the manuscript can be accepted.

- 1) P. 3, line 93. The authors describe the design motif associated with the photocathode and point out the requirement of no direct contact between the catalyst and the FTO substrate. However, this statement is in sharp contrast with the results obtained by SEM-EDS where a rather homogeneous distribution of Fe is apparent (Fig. 2). Please clarify this issue.
- 2) P. 4, line 149. The authors comment on a large photocurrent density of 181 $\mu\text{A}/\text{cm}^2$, but from Fig. 3b it is clear that the measured current does not reach zero upon removal of the light input. The photocurrent must be determined as the difference between the current measured under light subtracted by the value under dark (dark current). Thus, the photocurrent value provided should be adjusted to a slightly lower value in order to account for this evidence.
- 3) P. 4 and after. The manuscript would benefit from inclusion of additional characterization data related to the novel photocathode. Particularly, photoaction spectra should be measured and IPCE data provided. From these data, internal quantum efficiency (IQE or APCE) can be then extracted by taking into account the absorption of the photoelectrode. Finally, knowing the injection yield and the APCE, the charge collection efficiency can be estimated. The resulting value can be highly relevant to discuss about the improved charge transport arising from the presence of the polymer redox-relay.
- 4) P. 6, line 191. The transient absorption studies are well conducted and speak in favor of the active role of the polymer matrix in enhancing charge transport. I am wondering, however, whether the authors consider measuring the transient absorption in the presence of the hydrogenase in order to follow the electron transfer from the reduced polymer to the catalyst and thus confirm that this process “is not limiting the device’s photocurrent”.
- 5) Partly connected to the previous point. I am wondering why the authors limit their analysis as a function of the pH to a very narrow range (6-7.5). Is it related to electrode/catalyst instability? What about shifting to acidic pH where the HER should become easier?

Reviewer #2 (Remarks to the Author):

The manuscript reports an original biohybrid strategy for overall solar-driven water splitting in photoelectrochemical cells. It focuses on the development of an efficient dye-sensitized photocathode based on a [FeFe]-hydrogenase as proton reduction catalyst. The integration of the redox-active polymer PolyV in the electrode architecture proved to be key to this strategy; it resulted in an exceptional increase in performance as well as stability, enabling to record unprecedented STH efficiencies for a dye-sensitized photocathode in a bias-free tandem cell configuration. Kudos to the authors for this very nice piece of work! This thorough study, based on carefully performed and clearly described experiments, is of broad and general interest to the community and clearly deserves to be published in Nature Communications. Minor comments and questions are outlined below.

- References related to the preparation of the nanoporous NiO films are missing (page 10, line 322). Did the authors use a specific procedure to increase the size of the pores in order to facilitate the penetration of the polymer and the enzyme (which looks optimal from the SEM measurements), as well as the diffusion of the buffer electrolyte throughout the electrode structure?
- It is stated page 10, line 327 that the optimized PolyV concentration is 1.5 mg.L⁻¹, which is not so clear on Figure 3a... The plateau is rather reached at a concentration of 3 mg.L⁻¹...
- In Figure 1, the blue arrow depicting the electron transfer from NiO to the sensitizer is upside down.
- Page 11, line 384: « The Faradaic efficiency for H₂ and O₂ was calculated with equation (1). The amount of enzyme in this calculation was based on how much enzyme was deposited on the electrode. » ?? The Faradaic efficiency does not depend on the amount of catalyst present at the surface of the electrode.

Reviewer #3 (Remarks to the Author):

This work represents a real breakthrough in the field, with the design of a water-splitting device based on an innovative dye-sensitized photocathode with a biocatalyst displaying unprecedented performance. This significant result is the fruit of the collaboration of three groups sharing their respective expertise and innovative findings accumulated over the last few years. The experiments are well performed and the data well analyzed. The manuscript is very clear and is adapted to the wide readership of the journal. I would recommend the publication of this manuscript after my unique, important comment is addressed.

Page 5, lines 164-70: How the remaining HER activity of the enzyme after the 5 hours is measured? Regarding the same experiments, how do the authors explain that the photocurrent remains continuous (Fig. 3c) while 22 % of the enzymes are no longer active? How do the authors explain the evolution of the faradaic yield in Fig. 3d during the first hour?

To ensure that this is not a latent time to generate an active iron-based catalyst, the authors could introduce a non-active iron complex (modification of the dithiolate ligand) into the active site of the hydrogenase and verify that the system is not active or reconstitute the enzyme with the FeS clusters (without the active site).

Page 4, line 136: conc PolyV, a plateau at 3 mg.mL⁻¹ in the figure and 2 mg.mL⁻¹ in the text

Response Letter

Dear reviewers,

Thank you very much for reviewing our manuscript of "*Embedding biocatalysts in a redox polymer enhances the performance of Dye-Sensitized Photocathodes in Bias-free Photoelectrochemical Water Splitting*" (NCOMMS-23-60841-T). All comments received are very helpful for us to improve the manuscript.

The manuscript now has been carefully revised according to all comments. The revised manuscript and supporting information with all changes highlighted in yellow are uploaded for review. Below we respond to the comments in detail, marked in blue.

Sincerely,

Haining Tian, on behalf of all co-authors

Response to Reviewer 1

Reviewer's comment: The manuscript by Tian and co-workers describes the preparation of a photocathode for hydrogen evolution based on sensitized NiO and incorporation of a hydrogenase in a redox-active viologen-based polymer. The photocathode is highly active towards photoelectrochemical hydrogen production reaching unprecedented photocurrent densities. Interestingly, the photocathode has been also combined with a BiVO₄ photoanode to perform bias-free photoelectrochemical water splitting. The manuscript is interesting and the results unprecedented in the field so that it may warrant publication. The following comments should be, however, addressed before the manuscript can be accepted.

1. P. 3, line 93. The authors describe the design motif associated with the photocathode and point out the requirement of no direct contact between the catalyst and the FTO substrate. However, this statement is in sharp contrast with the results obtained by SEM-EDS where a rather homogeneous distribution of Fe is apparent (Fig. 2). Please clarify this issue.

Response: We thank the reviewer for the comment. We don't think the hydrogenase can cross the NiO blocking layer as the blocking layer is very compact and prepared from a sputtering method. As the blocking layer is very thin and also the hydrogenase might fall to the cross-section when we cut the film for the SEM test, it is hard to clearly see the boundary between NiO blocking layer and hydrogenase.

2. P. 4, line 149. The authors comment on a large photocurrent density of 181 $\mu\text{A}/\text{cm}^2$, but from Fig. 3b it is clear that the measured current does not reach zero upon removal of the light input. The photocurrent must be determined as the difference between the current measured under

light subtracted by the value under dark (dark current). Thus, the photocurrent value provided should be adjusted to a slightly lower value in order to account for this evidence.

Response: We thank the reviewer for raising this important consideration. After the subtraction by the dark current, the photocurrent stated in the revised manuscript has been changed to $141\pm 17 \mu\text{A}\cdot\text{cm}^{-2}$, which is still among the best photocurrent obtained from dye-sensitized NiO photocathode for fuel production. The corresponding changes are made on Page 1, 2 and 4.

3. P. 4 and after. The manuscript would benefit from inclusion of additional characterization data related to the novel photocathode. Particularly, photoaction spectra should be measured and IPCE data provided. From these data, internal quantum efficiency (IQE or APCE) can be then extracted by taking into account the absorption of the photoelectrode. Finally, knowing the injection yield and the APCE, the charge collection efficiency can be estimated. The resulting value can be highly relevant to discuss about the improved charge transport arising from the presence of the polymer redox-relay.

Response: We appreciate the suggestion. We have measured the photocurrent of the photocathode under LED lamps with wavelength at 470 nm, 590 nm and 630 nm (Supplementary Fig. 9). The IPCE and APCE data is provided for each sample with or without PolyV, showing the assisted charge transport by the redox polymer.

4. P. 6, line 191. The transient absorption studies are well conducted and speak in favor of the active role of the polymer matrix in enhancing charge transport. I am wondering, however, whether the authors consider measuring the transient absorption in the presence of the hydrogenase in order to follow the electron transfer from the reduced polymer to the catalyst and thus confirm that this process “is not limiting the device’s photocurrent”.

Response: Thank the reviewer for this suggestion. In our previous work, we have demonstrated the fast intermolecular electron transfer between the redox polymer and the hydrogenase by cyclic voltammetry (*Nat. Catal.* 2021, 4, 251-258). In addition, the pH independency of the photocurrent (Supplementary Fig. 10) also indicates that the electron transfer between the polymer and the enzyme is not the limiting step in the overall electron transfer pathway. In the fs-TA experiment, the reduced polymer was observed; however, the signal was weak, posing challenges for conducting subsequent TA experiments in the presence of hydrogenase. Therefore, in this work we are only able to monitor the electron extraction from the dye to PolyV.

5. Partly connected to the previous point. I am wondering why the authors limit their analysis as a function of the pH to a very narrow range (6-7.5). Is it related to electrode/catalyst instability? What about shifting to acidic pH where the HER should become easier?

Response: We agree with the reviewer that the environment with acidic pH is preferable for HER. In this work, the catalysis experiments were conducted in a neutral pH range (6-7.5), due to the best stability and activity of the enzyme in a neutral environment. It is possible to obtain high HER currents at $\text{pH} < 6$, but at the expense of a limited stability.

Response to Reviewer 2

Reviewer's comment: The manuscript reports an original biohybrid strategy for overall solar-driven water splitting in photoelectrochemical cells. It focuses on the development of an efficient dye-sensitized photocathode based on a [FeFe]-hydrogenase as proton reduction catalyst. The integration of the redox-active polymer PolyV in the electrode architecture proved to be key to this strategy; it resulted in an exceptional increase in performance as well as stability, enabling to record unprecedented STH efficiencies for a dye-sensitized photocathode in a bias-free tandem cell configuration. Kudos to the authors for this very nice piece of work! This thorough study, based on carefully performed and clearly described experiments, is of broad and general interest to the community and clearly deserves to be published in Nature Communications. Minor comments and questions are outlined below.

1. References related to the preparation of the nanoporous NiO films are missing (page 10, line 322). Did the authors use a specific procedure to increase the size of the pores in order to facilitate the penetration of the polymer and the enzyme (which looks optimal from the SEM measurements), as well as the diffusion of the buffer electrolyte throughout the electrode structure?

Response: We appreciate the reviewer for raising the question. One reference (*Phys. Chem. Chem. Phys.* 2018, 20, 36-40) with detailed information has been added to the method description for the preparation of PB6-sensitized mesoporous NiO films. We followed the method to prepare NiO films, with optimization for NiO thickness in order to achieve optimal light absorption and resistance of the photocathode.

2. It is stated page 10, line 327 that the optimized PolyV concentration is 1.5 mg.L^{-1} , which is not so clear on Figure 3a... The plateau is rather reached at a concentration of 3 mg.L^{-1} ...

Response: We apologize for the confusion. The optimal concentration of PolyV is 3 mg ml^{-1} . The value of 1.5 mg ml^{-1} is the PolyV concentration comes from the earlier stages, before optimization. The experimental part has been corrected to 3 mg ml^{-1} .

3. In Figure 1, the blue arrow depicting the electron transfer from NiO to the sensitizer is upside down.

Response: We thank the reviewer for pointing out this mistake. We have corrected Fig. 1 in the revised manuscript.

4. Page 11, line 384: « The Faradaic efficiency for H_2 and O_2 was calculated with equation (1). The amount of enzyme in this calculation was based on how much enzyme was deposited on

the electrode. » ?? The Faradaic efficiency does not depend on the amount of catalyst present at the surface of the electrode.

Response: The reviewer is of course absolutely correct. Faradaic efficiency represents the gas amount generated by certain number of electrons. It is not related to the catalyst amount at the electrode surface. The sentence has been deleted for precise description.

Response to Reviewer 3

Reviewer's comment: This work represents a real breakthrough in the field, with the design of a water-splitting device based on an innovative dye-sensitized photocathode with a biocatalyst displaying unprecedented performance. This significant result is the fruit of the collaboration of three groups sharing their respective expertise and innovative findings accumulated over the last few years. The experiments are well performed and the data well analyzed. The manuscript is very clear and is adapted to the wide readership of the journal. I would recommend the publication of this manuscript after my unique, important comment is addressed.

1. Page 5, lines 164-70: How the remaining HER activity of the enzyme after the 5 hours is measured?

Response: We apologize for this omission in the original submission and thank the reviewer for spotting it. We added the method description on Page 12 in the revised manuscript. *“The activity of the enzyme on the electrode surface before and after the 5-hour reaction was analyzed by putting the electrode sample into 0.1 M phosphate buffer (pH=6.8) containing 10 mM methyl viologen in a gas-tight vial and measuring the generated H₂ amount in GC (PerkinElmer Clarus 500) after incubation at 30 °C for 15 min following injection of sodium dithionite (final concentration 100 mM). The final volume of the solution was 400 mL. The percentage of residual enzyme activity after the reaction was obtained by the ratio of H₂ production.”*

2. Regarding the same experiments, how do the authors explain that the photocurrent remains continuous (Fig. 3c) while 22 % of the enzymes are no longer active?

Response: We thank the reviewer for this question. The result proves that the enzyme is active enough to sustain the photocurrent, even with a small amount. The photocurrent is mainly limited by the charge recombination at NiO surface.

3. How do the authors explain the evolution of the faradaic yield in Fig. 3d during the first hour? To ensure that this is not a latent time to generate an active iron-based catalyst, the authors could introduce a non-active iron complex (modification of the dithiolate ligand) into the active site of the hydrogenase and verify that the system is not active or reconstitute the enzyme with the FeS clusters (without the active site).

Response: We thank the reviewer for this suggestion. The decrease in the Faradaic efficiency for the NiO|PB6|H₂ase sample is due to the deactivation of the enzyme probably from enzyme detachment, and the slight increase in the Faradaic yield for the NiO|PB6|PolyV|H₂ase electrode could be ascribed to the rearrangement of PolyV and hydrogenase on NiO surface upon light illumination and catalysis to reach the equilibrium during the first hour. Considering

the magnitude of the current, it is very difficult to rationalize these findings without the high activity of CrHydA1

4. Page 4, line 136: conc PolyV, a plateau at 3 mg.mL⁻¹ in the figure and 2 mg.mL⁻¹ in the text.

Response: We thank the reviewer for spotting this mistake. This has been corrected to 3 mg.mL⁻¹ in the text.

REVIEWERS' COMMENTS

Reviewer #1 (Remarks to the Author):

Most of the comments raised have been adequately addressed. The manuscript can be publishable in its current form.

Reviewer #2 (Remarks to the Author):

All the reviewers' comments have been satisfactorily taken into account in the revised version; I therefore recommend publication of this manuscript.

Reviewer #3 (Remarks to the Author):

The authors have taken into account my comments and I'm pleased with their answers. I fully recommend the publication of his manuscript as it is.